# Mathematical Modeling of Rhesus Cytomegalovirus Transplacental Transmission in Seronegative Rhesus Macaques

**DOI:** 10.3390/v15102040

**Published:** 2023-10-01

**Authors:** Yishu Gong, Matilda Moström, Claire Otero, Sarah Valencia, Alice F. Tarantal, Amitinder Kaur, Sallie R. Permar, Cliburn Chan

**Affiliations:** 1Department of Mathematics, Duke University, Durham, NC 27710, USA; yishugong@hsph.harvard.edu; 2Department of Immunology, Tulane National Primate Research Center, Covington, LA 70433, USA; mmostrom@tulane.edu (M.M.); akaur@tulane.edu (A.K.); 3Department of Pathology, Duke University, Durham, NC 27710, USA; clo4001@med.cornell.edu; 4Duke Human Vaccine Institute, Duke University Medical Center, Durham, NC 27710, USA; sarah.valencia@duke.edu; 5Department of Pediatrics, School of Medicine, California National Primate Research Center, UC Davis, Davis, CA 95616, USA; aftarantal@ucdavis.edu; 6Department of Pediatrics, Joan & Weill Cornell Medicine, New York City, NY 10065, USA; sallie.permar@med.cornell.edu; 7Department of Biostatistics and Bioinformatics, Duke University, Durham, NC 27710, USA; 8Center for Human Systems Immunology, Duke University, Durham, NC 27710, USA

**Keywords:** cytomegalovirus, placental transmission, rhesus macaques

## Abstract

Approximately 0.7% of infants are born with congenital cytomegalovirus (CMV), making it the most common congenital infection. About 1 in 5 congenitally infected babies will suffer long-term sequelae, including sensorineural deafness, intellectual disability, and epilepsy. CMV infection is highly species-dependent, and the rhesus CMV (RhCMV) infection of rhesus monkey fetuses is the only animal model that replicates essential features of congenital CMV (cCMV) infection in humans, including placental transmission, fetal disease, and fetal loss. Using experimental data from RhCMV seronegative rhesus macaques inoculated with RhCMV in the late first to early second trimesters of pregnancy, we built and calibrated a mathematical model for the placental transmission of CMV. The model was then used to study the effect of the timing of inoculation, maternal immune suppression, and hyper-immune globulin infusion on the risk of placental transmission in the context of primary and reactivated chronic maternal CMV infection.

## 1. Introduction

Cytomegalovirus, or CMV, is the most common cause of congenital infection in humans and results in significant childhood morbidity. Congenital CMV (cCMV) occurs in 0.7% of all pregnancies [1] and is a major cause of childhood hearing loss and neurodevelopmental delay. In 2000, the Institute of Medicine prioritized the development of a CMV vaccine as its highest concern [2]. Despite numerous vaccine candidates undergoing clinical trials, none have thus far achieved the required efficacy levels for licensing. Consequently, there exists an immediate need to comprehend the factors governing transmission to facilitate vaccine development and passive immunization strategies. It is crucial to note that CMV infections are species-specific, resulting in pathology specific to each species. Only non-human primate (NHP) CMV models have been able to accurately replicate the neurological deficits observed in cCMV [3]. The rhesus macaque CMV (RhCMV) model has been instrumental in revealing the critical roles of maternal CD4+ T cells and pre-existing antibodies in preventing fetal transmission and associated diseases [3,4]. Previously, we utilized secondary statistical analysis to identify antibody variables that predicted the efficacy of the gB/MF59 vaccine in humans [5]. However, gaining an understanding of the immune mechanisms capable of preventing CMV transmission and cCMV required meticulously designed studies. The complexity of employing logistically intricate experimental NHP models such as RhCMV to investigate the determinants of cCMV transmission and disease presented a substantial challenge in our quest to understand and develop effective therapies for preventing this disease. The study of primary RhCMV infection during pregnancy proved especially formidable, given that RhCMV is prevalent in rhesus colonies. Stringent measures had to be implemented to house seronegative animals in separate housing areas for such investigations. In our pursuit of a deeper understanding of these and other experimental findings, we developed a mechanistic model for RhCMV transmission from mother to fetus. This model was calibrated using data from our experimental studies involving primary RhCMV infection in seronegative rhesus dams, which, to the best of our knowledge, encompassed the largest RhCMV challenge experiment ever conducted in immunocompetent seronegative rhesus dams.

There have been a number of published mathematical models of within-host CMV infection [6,7,8,9]. However, none of these prior studies directly addressed the modeling of the CMV transmission process from maternal blood across the placenta to the fetus. Our particular interest lay in the development of a model capable of estimating the probability of placental transmission using solely maternal viral load measurements. Furthermore, we aimed to employ this model for simulating therapeutic interventions at preventing congenital infection, thus providing valuable insights for vaccine development.

In this study, we developed a mathematical model to investigate CMV transmission to explore the impact of maternal immunity, placental factors, and immune interventions such as CD4+ T cell depletion or hyperimmune globulin (HIG) therapy on the number of transmitted viruses, the timing of infection, and the overall risk of congenital infection in the fetus. Our transmission model generated predictions that aligned with experimental results and clinical trial observations, offering a concise explanation for how the risk of transmission was influenced by the timing of primary infection and providing testable hypotheses. We firmly believe that this model can contribute to the future design of studies related to human CMV (HCMV) and aid in optimizing intervention experiments in NHPs to maximize the utility of limited sample sizes.

## 2. Materials and Methods

### 2.1. Experimental Procedure

The model was calibrated using data obtained from primary RhCMV infection of rhesus macaque dams [3,4]. In these experiments, pregnant RhCMV seronegative rhesus macaques were categorized into two groups: an immunocompetent group (*n* = 15) and an immune-suppressed group (*n* = 9). The immune-suppressed group underwent CD4+ T cell depletion through anti-CD4 monoclonal antibody infusion before inoculation. Macaques in both groups received RhCMV infection through intravenous injection. The inoculation with RhCMV was administered to both groups during the late first to early second trimester of pregnancy. Subsequently, RhCMV viral loads in maternal plasma and amniotic fluid were regularly monitored over time using quantitative PCR (qPCR). The lower limit of detection for the qPCR assay was set to 1–10 copies of RhCMV DNA in each PCR reaction. To reduce the risk of false positives from inadvertent contamination, when the qPCR results approached this lower limit of detection, PCR reactions were deemed positive if at least 2 out of 6 replicates in the plasma or 2 out of 12 replicates in the amniotic fluid showed positive results. Consequently, six out of fifteen immunocompetent animals displayed detectable RhCMV DNA in their amniotic fluid [3]. All nine CD4+ T-cell-depleted animals exhibited detectable RhCMV DNA in their amniotic fluid, signifying 100% placental transmission [3,4]. The experimental measurements are depicted in Figure 1. It is worth noting that five data points for the control dams (145-97, 174-97, 369-09, GM04, and HD79) have been previously published in [4], and three data points for the CD4+ T-cell-depleted dams (223-98, 251-05, 273-98) can be found in [3]. The data for the remaining dams were derived from new experiments.

### 2.2. Model Structure

In our study, we modeled CMV dynamics in three key compartments: the dam, placenta, and fetus. Within the maternal compartment, we depicted complex interactions involving the virus, immune cells, and cells within the maternal bloodstream using a system of ordinary differential equations (ODEs). This maternal compartment is interconnected with the placental compartment, where we emulated the transmission of CMV from mother to fetus employing a diffusion process with decay. The placental compartment, in turn, is linked to the fetal compartment as it serves as the primary source of the virus entering the fetus. Within the fetal compartment, we focused on virus entry and progression within the fetus, employing stochastic models to account for the inherent uncertainties associated with smaller populations.

#### 2.2.1. Maternal Compartment

We employed a model originally developed for CMV infection in kidney transplant recipients undergoing immune suppression therapy [6] to describe immune and viral dynamics in the dam. This model expanded upon the conventional virus dynamics model [10] by introducing an additional term that controlled the degree of immune suppression. The structural representation of this model is depicted in Figure 2, and it was expressed as a system of ODEs:(1)$$\begin{array}{c} \begin{array}{cc} dV/dt & =n\delta R_{I}-cV-fkR_{S}V,\\ dE/dt & =\left(1-\epsilon_{S}\right)\left(\lambda_{E}\left(1-\frac{E}{e}\right)E+\rho V\right),\\ dR_{I}/dt & =kR_{S}V-\delta R_{I}-\left(1-\epsilon_{S}\right)mER_{I}+\alpha_{0}R_{L}-\kappa R_{I},\\ dR_{S}/dt & =\lambda \left(1-\frac{R_{S}}{r_{S}}\right)R_{S}-kR_{S}V,\\ dR_{L}/dt & =\lambda \left(1-\frac{R_{L}}{r_{L}}\right)R_{L}+\kappa R_{I}. \end{array} \end{array}$$

In the system of ODEs (Equation 1), *V* denotes viral load (free virus) per µL-blood; *E* denotes virus-specific immune effector cells per µL-blood; and $R_{I}$, $R_{S}$, and $R_{L}$ denote actively infected, susceptible cells, and latently infected cells per µL-blood, respectively. A description of all model parameters is given in Table 1. Since we used the original paper’s parameter for initial guesses, our fitted values are within a reasonable range. Note that the original paper [6] delves into pertinent clinical measurements in Section 3, while Section 4 is dedicated to the approximation of the parameters.

The steady-state analysis is algebraically complicated because of the many nonlinearities, including three logistic growth functions; hence, here we provide a heuristic argument. Assuming reasonable parameter values, we can derive several insights about equilibrium states and consequently deduce the existence of a solitary biologically relevant and stable equilibrium state. The trivial equilibrium state, where all variables equals zero in Equation (Equation 1), is inherently unstable. Once an infection takes root, in the ODE system, eradication becomes implausible due to the self-regeneration of latent cells, labeled as $R_{L}$. This mandates the consideration of a non-zero solution—one that remains bounded. Under the assumption that the rates of reversion for $R_{S}$ and $R_{L}$ ($\lambda =1\cdot 10^{-3}$) are notably lower than the growth rate of E(ρ=5)—a plausible conjecture given their variance of three orders of magnitude—the virus is inherently curbed from surging infinitely. This restraint arises from the controlled escalation of effector cells *E* at a rate of ρV, effectively maintaining a leash on the progression of $R_{I}$ (and, by extension, *V*) through its regulation by *E*. Given the instability of the trivial equilibrium state and the boundedness of the alternative solution, it means that the bounded solution must be stable, which is also shown in the original paper [6].

#### 2.2.2. Placental Compartment

We modeled the transmission of CMV across the placenta as a diffusion process with decay. While physical diffusion might not serve as a precise physical model for CMV in the placenta, we considered it as a phenomenological representation intended to capture the overall dynamics of placental transmission at a coarse-grained level. Similar partial differential equation (PDE) models have demonstrated their capability to accurately represent the dynamics of HIV passage through the urogenital mucosal epithelium [11]. For simplicity, we assumed homogeneous dynamics in the radial directions and represented diffusion through the placenta depth-wise as a 1D diffusion process, extending from the maternal blood lake to fetal circulation. This approach yielded:(2)$$\frac{\partial Q}{\partial t}=D\frac{\partial Q^{2}}{\partial x^{2}}-\mu Q.$$
with boundary conditions:(3)Q=0,x=0,t≥0,
(4)Q=V(t),x=l,t≥0,
and initial condition:(5)Q=0,0<x<l,t=0.

Here, *Q* denotes the concentration of CMV. Constant *D* denotes the diffusion coefficient. Constant μ denotes the death rate of the CMV. V(t) refers to the maternal viral load estimated by piecewise cubic hermite interpolating polynomial (PCHIP) or ODE model fits. The boundary conditions assume that the virus concentration at the maternal–placental interface is the same as the viral concentration in maternal blood and that once a virus crosses the placenta into the fetus it is carried away rapidly by the fetal blood flow.

With simplified versions of these boundary conditions, such as assuming a constant maternal viral concentration or approximating it with a simple polynomial, Equations (Equation 2) through (Equation 5) could be solved analytically using Fourier series to derive a closed-form solution. This analytical solution played a crucial role in verifying the accuracy of numerical simulations, particularly when dealing with more intricate maternal viral load dynamics.

The number of viruses that enter the fetus, denoted by *C*, is estimated by:(6)$$C=\int_{0}^{T}D\frac{\partial Q}{\partial x}{|}_{x=0}G\left(t\right)dt,$$
where G(t) refers to the growth curve of the placenta throughout pregnancy. Here we assume that G(t) has the form of a logistic curve in the range where we have data points and has a linear expression that connects to the origin in early pregnancy:(7)$$G\left(t\right)=\left\{\begin{array}{c} \frac{a}{1+e^{-(b(t-c))}},\mathrm{if}\;t>40;\\ g\cdot t,\mathrm{if}\;0\leq t\leq 40 \end{array}\right.$$

By treating the viral flux
(8)$$\lambda \left(t\right):=D\frac{\partial Q}{\partial x}\left(0,t\right)G\left(t\right)$$
as the rate for an inhomogeneous Poisson point process, we can obtain the time that each virus arrives in the fetal compartment.

#### 2.2.3. Fetal Compartment

We modeled the arrival and subsequent fate of each CMV virion that entered the fetus utilizing a stochastic process. This approach allowed us to accurately depict the dynamics of small populations. For the purpose of our modeling, we assumed a constant number of susceptible cells, as we specifically focused on early infection dynamics in the fetus. After the virus entered a susceptible cell, we considered two stochastic models for the subsequent dynamics, one that included latent viral infection stages and one that did not:

Model 1: Susceptible cell with latent stage:(9)$$\begin{array}{c} \begin{array}{cc} S+V\overset{}{\to }L & \;\mathrm{with}\;c_{1}=\beta SV,\\ L\overset{}{\to }\phi & \;\mathrm{with}\;c_{2}=d,\\ L\overset{}{\to }I & \;\mathrm{with}\;c_{3}=\alpha ,\\ I\overset{}{\to }\phi & \;\mathrm{with}\;c_{4}=d,\\ I\overset{}{\to }I+V & \;\mathrm{with}\;c_{5}=\omega ,\\ V\overset{}{\to }\phi & \;\mathrm{with}\;c_{6}=\gamma . \end{array} \end{array}$$
Model 1 corresponds to a target-cell limitation model given in [12], which has a reproductive number $R_{0}$ given by:(10)$$R_{0}=\frac{\beta S_{0}\omega }{cd(1+\frac{\gamma }{\alpha })}=1.02.$$Model 2: Susceptible cell without latent stage:(11)$$\begin{array}{c} \begin{array}{cc} S+V\overset{}{\to }I & \;\mathrm{with}\;c_{1}=\beta SV,\\ I\overset{}{\to }\phi & \;\mathrm{with}\;c_{2}=d,\\ I\overset{}{\to }I+V & \;\mathrm{with}\;c_{3}=\omega ,\\ V\overset{}{\to }\phi & \;\mathrm{with}\;c_{4}=\gamma . \end{array} \end{array}$$
Model 2 corresponds to a target infection virus model given in [13], which has a reproductive number $R_{0}$ given by:(12)$$R_{0}=\frac{\beta S_{0}\omega }{cd(1+\frac{\beta S_{0}}{c})}$$

Viral latency only occurs in some cell types, including myeloid cells. Since the initial transmission event in the model is via the blood, and the first cells infected may be myeloid cells, we assume that both life cycles (Model 1 and Model 2) are possible. For each virion, we selected between Model 1 and Model 2 based on the following procedure: We assumed a ratio of r:(1−r) between the susceptible cells that enter the latent stage and the susceptible cells that become infected cells. Then, we drew a uniform random variable *s* from the range 0 to 1. If s<r, we implemented Model 1 as defined in Equation (Equation 9). Conversely, if s>r, we implemented Model 2 as defined in Equation (Equation 11) (in our sample runs, we used r=0.5 as the ratio). Since we assumed that the number of susceptible cells remained constant throughout the period of interest, we did not account for the production and death of susceptible cells. Specifically, *S* was set to $S_{0}$, and thus $\beta S=\beta S_{0}$ remained a constant value of 0.0012 (virions·day)${}^{-1}$. The true value of $R_{0}$ is highly uncertain, and we also performed sensitivity analysis where we explored scenarios in which each virion exclusively adhered to either Model 1 or Model 2 (as detailed in Section 3.5).

Upon determining the arrival time of each virus, we employed the Gillespie algorithm [14] to simulate viral dynamics within the fetus. We defined a fetus as infected once the viral load exceeded a predetermined threshold. By repeatedly conducting these simulations, we derived a distribution for the time of infection and the probability of transplacental transmission.

### 2.3. Model Calibration

#### 2.3.1. Maternal Compartment

We used the parameters for HCMV infection given in Kepler et al [6] as initial guesses but kept half-life of virions ($t_{H}$), equilibrium level of CMV-specific effector cells ($\tilde{E}$), and equilibrium level of virions ($\tilde{V}$) fixed. Justification for these parameter values with respect to clinical measurements and approximations used can be found in [6]. In addition, the initial value for the ODE fit used was $V\left(0\right)=1\cdot 10^{-4},E\left(0\right)=0,R_{I}\left(0\right)=0,R_{S}\left(0\right)=400$, $R_{L}\left(0\right)=0$ per µL of blood. When we performed parameter estimation for immunocompetent rhesus macaques, we set level of immune suppression ($\epsilon_{S}$) to 0.

The remaining parameters for the system of ordinary differential equations were estimated via nonlinear mixed-effects regression, as implemented in the Monolix software (version 2020R2, Lixoft) [15]. Monolix uses stochastic approximation expectation maximization (SAEM) to perform parameter estimation [16] and can accommodate the use of viral loads below the limit of detection (left censoring). Population parameter estimates for both the immunocompetent and CD4+ T-cell-depleted groups described in the ODE system (Equation 1) are summarized in Table 1. As expected, the fitted values for rhesus macaques are not identical to those estimated for humans but also do not deviate excessively. We also present the interpolation and ODE fits for maternal viral loads in Figure 3 and Figure 4.

#### 2.3.2. Placental Compartment

We first estimated the growth curve for the placenta given by Equation (Equation 7). Using the data from [17] for pregnant rhesus macaques, we fitted a logistic function to obtain Figure 5 with a= 132,434, b=0.028, c=90, and g=667.

For boundary conditions on the maternal–placental interface, we needed to estimate a continuous function from discrete experimental measures of viral loads over time. As all immunocompetent rhesus macaques survived until fetal tissue harvests via hysterotomy, we fitted the experimental measures directly using piecewise cubic hermite interpolating polynomial (PCHIP) interpolation [18]. Unlike the more commonly used cubic spline interpolation, PCHIP avoids overshooting and can accurately connect the flat regions without creating artificial oscillations. For CD4+ T-cell-depleted rhesus macaques, 5 out of 9 were euthanized at peak viremia due to disseminated RhCMV; thus, we needed to extrapolate using the ODE model (Equation 1) proposed by [6] to obtain a smooth trajectory for each animal. In addition to individual maternal viral dynamics, population fits for immunocompetent and CD4+ T-cell-depleted dam groups were obtained using (Equation 1) with a nonlinear mixed effects model.

For the partial differential Equation (Equation 2), we used the diffusion coefficient of HIV through stroma as our *D* [11], and we include a sensitivity analysis in Section 3.5. We observed from the experimental data that all fetal infections occurred within 2–4 weeks after the dams were infected with RhCMV. Using this finding, we approximated the distance *l* that the virus needs to travel. The diffusion coefficient determines the time it takes a solute to diffuse a given distance in a medium:$$t\approx \frac{l^{2}}{2\cdot D}.$$
Then we have:$$l\approx \sqrt{2\cdot t\cdot D}\approx 0.6\;\mathrm{mm}.$$
We also calibrated μ in (Equation 2) using the experimental data such that the probability of placental RhCMV transmission for immunocompetent rhesus macaques using stochastic simulation is between 30% to 40%, and we obtained μ=1 (day)${}^{-1}$.

#### 2.3.3. Fetal Compartment

For the stochastic simulation of the fate of CMV arrivals in the fetus given by Equations (Equation 9) and (Equation 11), all parameters are from [12] and are shown in Table 2.

We treated the viral flux at the placental–fetal boundary as an arrival rate of new virus particles into the fetus. By treating the arrivals as an inhomogeneous Poisson process, we obtained the arrival times for each virus. Figure 6 shows the viral flux functions and simulated arrival times corresponding to population fits of immunocompetent and CD4+ T-cell-depleted rhesus macaques.

### 2.4. Probability of Transplacental Transmission

CMV is not a very infectious virus [8], and the ratio of infectious virions to genomes in HCMV is believed to be in the range of 1:150 to 1:1000 [19]. We defined a fetus to be infected when the number of virions crossing the placenta exceeds a threshold *M* and calibrated this threshold for infection with experimental data so that the primary infection rate during the first trimester was approximately 40% for immunocompetent rhesus macaques and ≥99% for the CD4 T-cell depleted rhesus macaques. We found *M* = 500 copies.

Our simulations revealed that once this threshold was exceeded, the spontaneous clearance of the virus became highly improbable. To account for the uncertainty in this estimate, we also examined various threshold values and present the results of this sensitivity analysis in Figure 7.

Next, we estimated the time to transmission. We first estimated the the probability *p* that a single replicating virus, passing through the placenta, would result in a sustained infection directly from stochastic simulations.

With initial values $S_{0}=4\cdot 10^{8},L_{0}=0,I_{0}=0,V_{0}=1$, we ran the Gillespie algorithm $10^{6}$ times with two stopping criteria:No more viruses, latent cells, or infected cells are left (no persistent infection);Virus count has reached *M* copies (persistent infection).

We obtained p=0.08%. Then, if we consider that every virus can cause persistent infection independently, we can estimate the probability of infection if *N* viruses travel through the placenta as:(13)$$P=1-{\left(1-p\right)}^{N}.$$

As a result, we obtained a cumulative probability over time by calculating the number of viruses using (Equation 8):(14)$$P\left(t\right)=1-{\left(1-p\right)}^{N(t)},\;N\left(t\right)=\int_{0}^{t}\lambda \left(\tau \right)d\tau .$$

This is a heuristic estimate since there can be multiple lineages from different founder viruses growing at the same time, which could reduce the time to infection.

## 3. Simulation Results

### 3.1. Immune Suppression Increases the Likelihood of Placental Transmission

Maternal CD4+ T cells played a pivotal role in preventing severe cCMV disease, as demonstrated in the RhCMV placental transmission model [3], as well as in humans [20]. Our model echoed these findings, underscoring how the extent of immune suppression significantly influenced the likelihood of transplacental transmission. In our analysis of experimental data, we observed that the CD4+ T cell-depleted group of rhesus macaque dams exhibited notably higher plasma viral loads. Consequently, this resulted in a higher number of viruses entering the fetus during pregnancy. As anticipated, this led to an increased probability of transplacental transmission within the CD4+ T cell-depleted dam group.

In Figure 8, our calibrated model illustrates that the probability of RhCMV transplacental transmission in immunocompetent rhesus macaque dams was estimated to be 35%, while the probability of RhCMV transplacental transmission in CD4+ T cell-depleted rhesus macaque dams was predicted to be 100%.

### 3.2. Inoculation in Different Trimesters

In humans, it has been observed that the risk of cCMV following primary infection is 30–40% in the first and second trimesters but rises to 40–70% in the third trimester [21]. We observed this same phenomenon in our model simulation when we shifted the time of viral inoculation. Maternal infections during 0≤t<55 days, 55≤t<110 days, and t≥110 days correspond to the first, second, and third trimester in the rhesus macaque model (term 165±10 days); for modeling purposes, 166 days was used [17]. The model indicated that the increase in risk during later trimesters could be attributed to placental growth, which induced a larger viral flux at the placenta–fetus boundary. Consequently, this led to a correspondingly higher probability of transplacental transmission, as illustrated in Figure 9 and Figure 10. Note that the model can predict outcomes for infection at any gestational age as noted by the smooth curve in Figure 9 and Figure 10. The trimester risk predictions were averaged over the smoothed curves. In addition to the placental considerations, other maternal adaptations that could potentially influence the model encompass alterations in the concentration of immune effector cells within the maternal system, as well as changes in the count of susceptible cells for fetal infection. Drawing insights from [22], it emerges that the overall count of CD8+ cells—integral components of the effector T cell population within the maternal ODE model—undergoes minimal fluctuations. For the fetus, we assumed that the number of susceptible cells is effectively unlimited in the early stages of viral infection, so changes in total number of susceptible cells over fetal growth would not change model predictions. While we cannot rule out that other developmental changes, such as changes in the immune milieu, also play a role [23], the model provided a parsimonious explanation and suggests placental growth as a contributing factor to transmission risk.

### 3.3. Prevention of cCMV Infection with Hyperimmune Globulin Following Primary Maternal Infection

Postinfection hyperimmune globulin (HIG) infusion was shown to reduce rates of transplacental HCMV transmission in several nonrandomized small-scale clinical trials [24,25] and case-controlled studies [26,27]. However, in a large-scale, randomized, placebo-controlled trial [28], the use of HIG after primary maternal CMV infection showed no efficacy.

We extended (Equation 14) to evaluate the utility of HIG infusion at different times following primary maternal CMV infection at preventing cCMV infection:(15)$$\begin{array}{c} \begin{array}{cc} P(t) & =1-{\left(1-p\right)}^{N(t)},\;N\left(t\right)=\int_{0}^{t}\lambda \left(\tau \right)H\left(\tau \right)d\tau ,\\ H(\tau ) & =\left\{\begin{array}{cc} 1, & \mathrm{if}\;\tau <T_{\mathrm{infusion}}\\ (1-\epsilon_{\mathrm{eff}}), & \mathrm{if}\;\tau \geq T_{\mathrm{infusion}} \end{array}\right. \end{array} \end{array}$$
In the above equation, $T_{\mathrm{infusion}}$ refers to the time of HIG infusion, and $\epsilon_{\mathrm{eff}}$ refers to the assumed effectiveness of HIG treatment. This simple model assumes that HIG maintains the same efficacy for all time and thus overestimates the utility in the real world, where HIG concentrations in plasma would diminish between infusions. The reduction in predicted risk of fetal CMV infection with different time and effectiveness of HIG infusion for immunocompetent and CD4+ T-cell-depleted immune suppressed hosts is shown in Table 3 and Table 4, respectively. Our simulations indicated that HIG infusion for immunocompetent hosts had limited utility if administered later than two weeks following maternal infection. Moreover, HIG infusion would need to be highly effective, approaching 100%, to effectively prevent infection, particularly in CD4 T cell-depleted animals. Given that maternal CMV infection often presents as asymptomatic, diagnosis relies on seroconversion, which typically takes at least two weeks. This suggests that HIG delivery based on serologic diagnosis may not be sufficient to prevent transplacental CMV transmission.

### 3.4. Primary and Reactivated Chronic Maternal RhCMV Infection

It is difficult to study the reactivation of chronic infection in an experimental setting in immunocompetent animals; thus, we ran simulations using our mathematical model to fill in this gap in knowledge. We simulated reactivated chronic maternal RhCMV infection by running the system of ODEs in (Equation 1) for an extended period until the viral dynamics approached equilibrium. We then compared the transmission probabilities between primary and reactivated chronic infections in Figure 11 and Figure 12. For both immunocompetent and CD4+ T-cell-depleted rhesus macaques, the risk of cCMV following primary maternal infection significantly exceeded that of reactivated chronic infection, indicating the important role preexisting immunity to CMV plays in suppressing transmission.

### 3.5. Sensitivity Analysis

For sensitivity analysis of the stochastic simulation, we show in Figure 7 the effect of ratio *r* for viruses in the fetus with and without a latent stage, the threshold *M* defining fetal infection, and the diffusion coefficient *D* for virus in the placenta. As shown, a higher threshold *M* gives a lower infection rate at delivery. In addition, we observed that the infection rate is lower at delivery if the proportion of viruses that enter the latent stage is higher. This observation corresponds to our $R_{0}$ calculation from (Equation 10) and (Equation 12).

## 4. Discussion

The determinants of cCMV transmission are challenging to study experimentally, as only the rhesus model recapitulates the features of human cCMV transplacental transmission and fetal neurological disease. NHP experiments are labor-intensive and hence need to be designed to maximize the generation of new knowledge. This is even more true for studying transmission in primary infection—since RhCMV is endemic in primate colonies. Thus, measures are taken, such as housing seronegative animals in separated areas and avoiding mixing with conventional colony animals, which adds to the cost and complexity of such studies. For these reasons, the cohort of RhCMV seronegative dams used to calibrate the model is likely the largest such cohort ever studied in a challenge experiment. The mathematical model for the transplacental transmission of CMV we describe here can serve as an in silico experimental system, facilitating the generation and identification of the most promising hypotheses and therapeutic interventions for validation in RhCMV experiments.

Our mathematical model for transplacental transmission considers CMV dynamics in the dam, the placenta, and the fetus. Each of these three physical compartments has its own unique characteristics that require a different mathematical framework to adequately represent. We also describe how we link the dynamics across physical compartments by careful consideration of conditions at the boundaries. Finally, we consider the simple fact that the placenta is growing throughout pregnancy and incorporate this growth curve into the model framework. A surprising hypothesis generated from our model is that the increase in transmission rates across different trimesters observed in human studies can be parsimoniously explained as a consequence of the surface area of the placenta at the time of peak viral load. Of course, many other physiological changes occurring in the mother, placenta, and fetus over pregnancy may also affect transmission risk, such as changes in concentration of immune effector cells in the mother, transplacental trafficking of IgG vs. IgM, or change in number of cells susceptible to CMV infection in the developing fetus. We are only reporting that models can be useful for hypothesis generation—in this case, the novel hypothesis that placental growth itself could be an unsuspected determinant of the transmission risk across different stages of pregnancy.

We then calibrated the model using data from RhCMV experiments that compared the risks of transplacental RhCMV transmission for immunocompetent and CD4+ T-cell-depleted hosts. These experiments collected longitudinal measurements of the viral load in maternal plasma and amniotic fluid, allowing us to also estimate the time of transmission in cases where the fetus was infected. Using the calibrated model, we then simulated the dynamics of transplacental RhCMV transmission in several scenarios and compared the risks of transmission with reactivated chronic versus primary maternal infection, with primary infection at different stages of pregnancy, and with immune suppression prior to inoculation or immune augmentation by HIG infusion following primary maternal infection. For each scenario, the model predicts the probability of infection, as well as the most likely time window for congenital infection to occur, and an estimate of the number of potential founder viruses that cross the placenta.

It is natural to consider the extent to which the rhesus model is applicable to human pregnancies. A comparison of rhesus macaque and human development is provided in [29], and the authors conclude that key developmental processes are fundamentally conserved among humans and monkeys, although the timing of when these events occur is obviously different given the divergence of their body sizes and gestation periods. One limitation of the rhesus model is that the intravenous route of infection is different from the mucosal route from which humans acquire CMV. This is unavoidable as CMV is not very infectious, and mucosal infection routes lead to variable and low-level viremia. Also, because of the low efficiency of mucosal infection, a much larger cohort of seronegative dams would be needed. While direct inoculation is not ideal, it is the best feasible design available and leads to similar pathologies in the infected fetuses as observed in human congenital infections [30].

The mathematical model and the in silico simulation of transplacental transmission is a highly simplified representation of a complex biological system [31,32], and there are several limitations. The most serious limitation is that several of the parameters are estimated from the literature based on different experimental contexts—for example, studies in humans rather than rhesus macaques. The models may also be uncertain because the biology is at present poorly characterized—for example, the CMV life cycle in the fetus. We have attempted to mitigate these shortcomings by performing sensitivity analyses for parameters where there is most uncertainty. These parameter estimates will improve over time as experiments are refined—for example, our collaborators have now developed rhesus placental organoid model systems that can be studied, and we are currently analyzing the detailed immune profiling data of the immunocompetent dams described in this manuscript. Finally, the predicted risk of transplacental transmission from reactivated chronic RhCMV infection is not based on experimental data but a model for chronic infection based on long-run values after acute infection. However, we believe that the modeling framework captures the major determinants of RhCMV transmission and can recapitulate experimental and clinical observations. For example, the model predicts that the number of transmitted founder viruses is typically small, consistent with studies of viral genomics in infected fetuses [33]. The model also predicts that delayed infusion of HIG more than 2 weeks after the primary maternal infection has little effect in preventing cCMV transmission among immunocompetent hosts, consistent with the negative results of a recent clinical trial [34] where expectant mothers infected before 23 weeks’ gestation were given monthly HIG infusions starting from serologic diagnosis of primary CMV infection based on low avidity CMV-specific IgG responses, which take at least 2 weeks to develop.

While we were privileged to work with such extensive data from a challenging and highly translational primate model system, our framework can benefit from (1) better characterization of the maternal CMV-specific innate and adaptive immune responses, so we can more accurately model viral control; (2) concentration of maternal anti-CMV antibodies from transplacental transfer in the fetus; and (3) calibration of the model using measurement of placental viral loads at the time of near-term fetal tissue collection by hysterotomy. With additional experimental data, the model can be extended to include more mechanistic hypotheses, for example to model the effect of vaccination or specific immune mechanisms that serve as barriers to transplacental transmission that are currently represented by a simple linear decay. Similarly, the simple diffusion model would benefit from direct measurement of CMV phenomenological diffusion rates in placental tissue rather than using estimates from an HIV model, and more accurate models of placental anatomy. Hence, we believe that this model provides an encompassing framework for future mechanistic models that will be useful for understanding, developing, and mitigating risk in clinical trials of interventions to prevent cCMV infection.

## Figures and Tables

**Figure 1 viruses-15-02040-f001:**
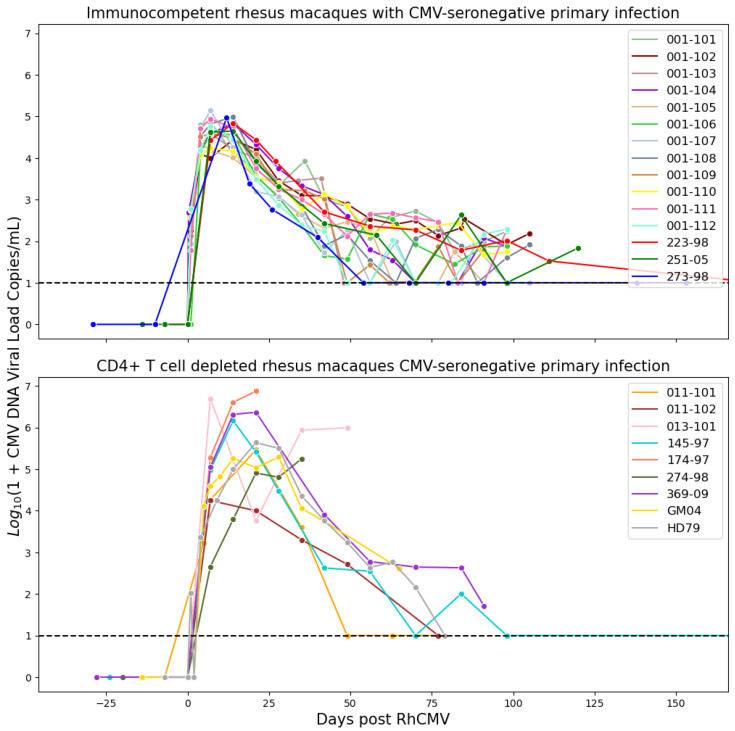
Plasma RhCMV viral loads in two groups of infected rhesus macaques (immunocompetent and CD4+ T-cell-depleted).

**Figure 2 viruses-15-02040-f002:**
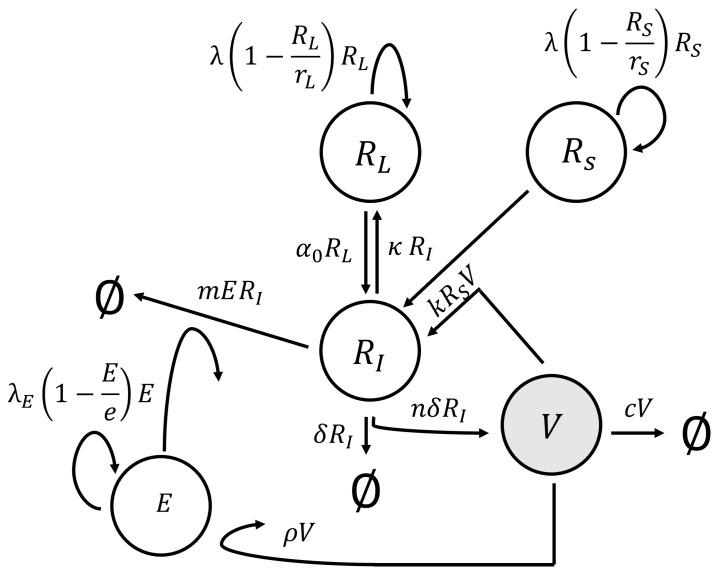
Schematic diagram of key processes in the model of HCMV infection.

**Figure 3 viruses-15-02040-f003:**
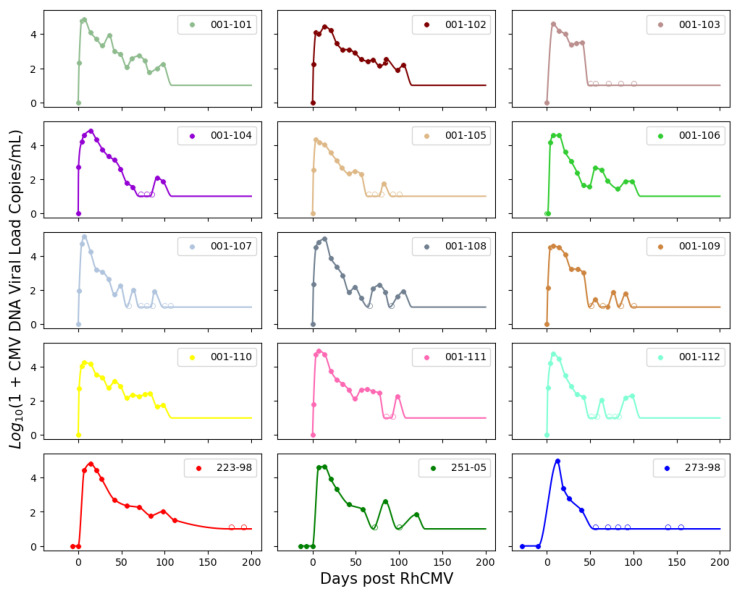
PCHIP interpolations for immunocompetent RhCMV seronegative rhesus macaque dams following primary RhCMV infection.

**Figure 4 viruses-15-02040-f004:**
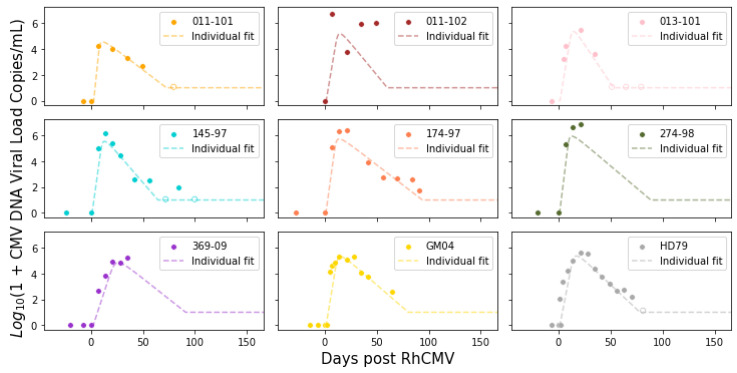
Fits from the system of ordinary differential equations for CD4+ T-cell-depleted RhCMV seronegative rhesus macaque dams following primary RhCMV infection. See fit diagnosis in Appendix A.

**Figure 5 viruses-15-02040-f005:**
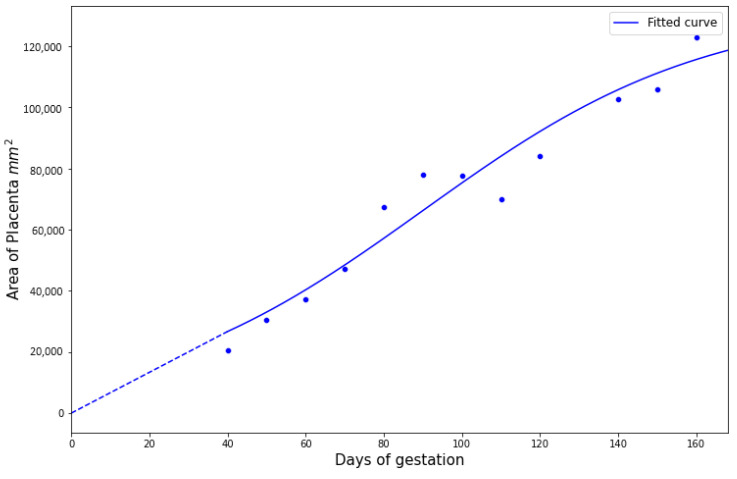
Growth function G(t) for placenta surface area over time with data from [17] (shown as points). The dotted line connects the first fitted value and the origin using linear interpolation.

**Figure 6 viruses-15-02040-f006:**
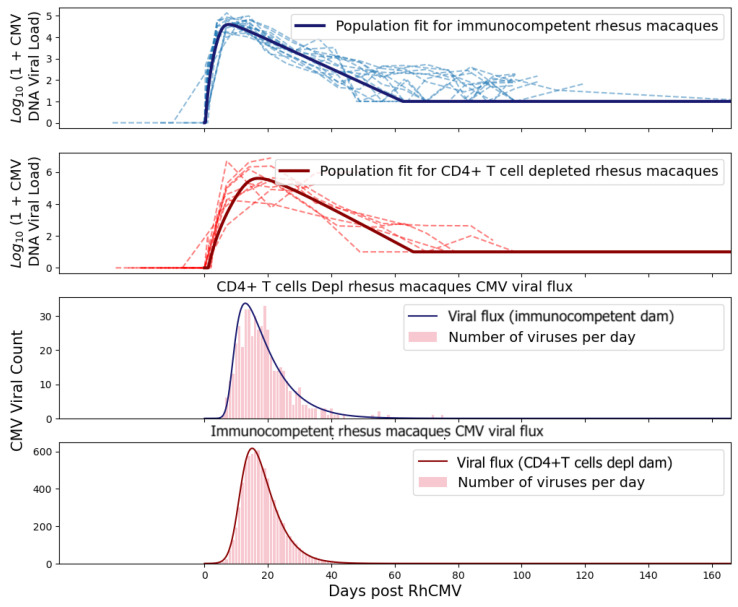
Inhomogeneous Poisson sampling from the viral flux. Note that the y-axis scaling for each plot is different. The dotted lines represent plasma RhCMV viral loads.

**Figure 7 viruses-15-02040-f007:**
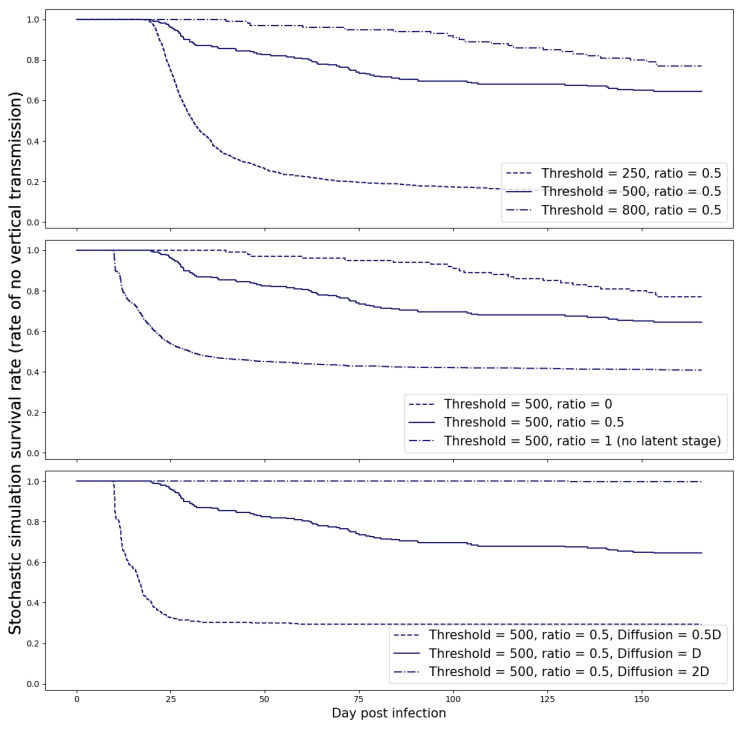
Sensitivity analysis for threshold *M*, ratio *r*, and diffusion coefficient *D* for stochastic simulation.

**Figure 8 viruses-15-02040-f008:**
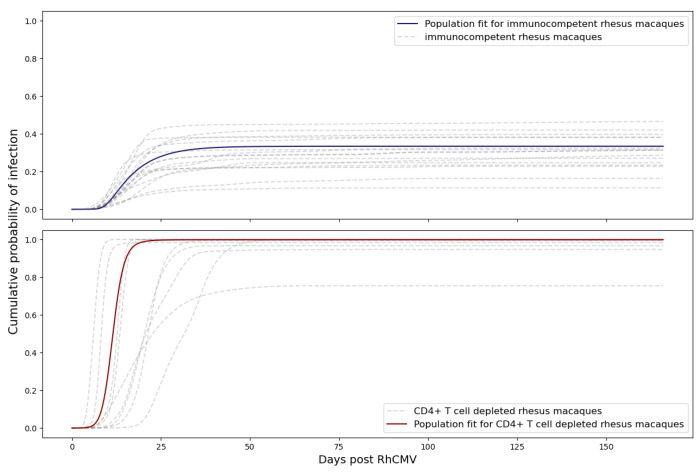
Probability of transplacental transmission of RhCMV following primary infection of immunocompetent and CD4+ T-cell-depleted dams. The blue line represents the result using the immunocompetent dam population fit as maternal viral dynamics. The gray dashed line represents each individual immunocompetent rhesus macaque. The red line refers to the result using the CD4+ T-cell-depleted macaque dam population fit to the system of ordinary differential equations for maternal viral dynamics. The gray dashed–dotted line refers to each individual CD4+ T-cell-depleted rhesus macaque. Graph generated taking M=500 copies and using (Equation 13).

**Figure 9 viruses-15-02040-f009:**
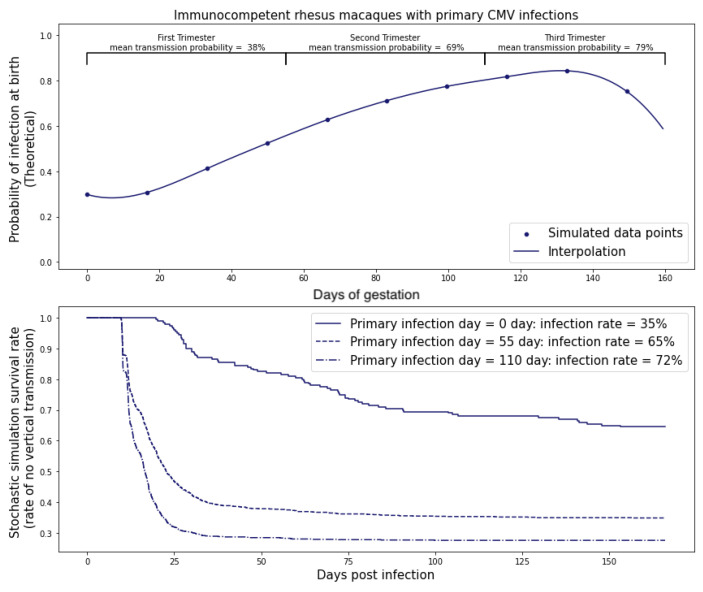
Probability of transplacental transmission of primary RhCMV infections during different trimesters of pregnancy in immunocompetent dams. The maternal dynamics come from the population fits of the immunocompetent group of rhesus macaques with starting time t=0 days (primary infection 1st trimester), time t=55 days (primary infection 2nd trimester), and t=110 days (primary infection 3rd trimester). Graph generated taking threshold M=500 copies.

**Figure 10 viruses-15-02040-f010:**
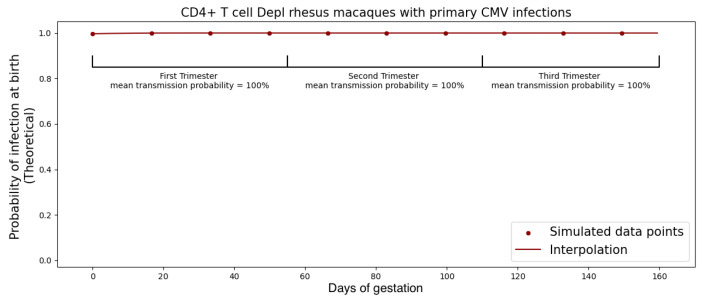
Probability of transplacental RhCMV transmission after primary maternal infections during different trimesters of pregnancy in CD4+ T-cell-depleted RhCMV seronegative dams. All rhesus macaques are infected 14 days post-inoculation since there are over *M* = 500 copies of viruses that cross the placenta in the same day.

**Figure 11 viruses-15-02040-f011:**
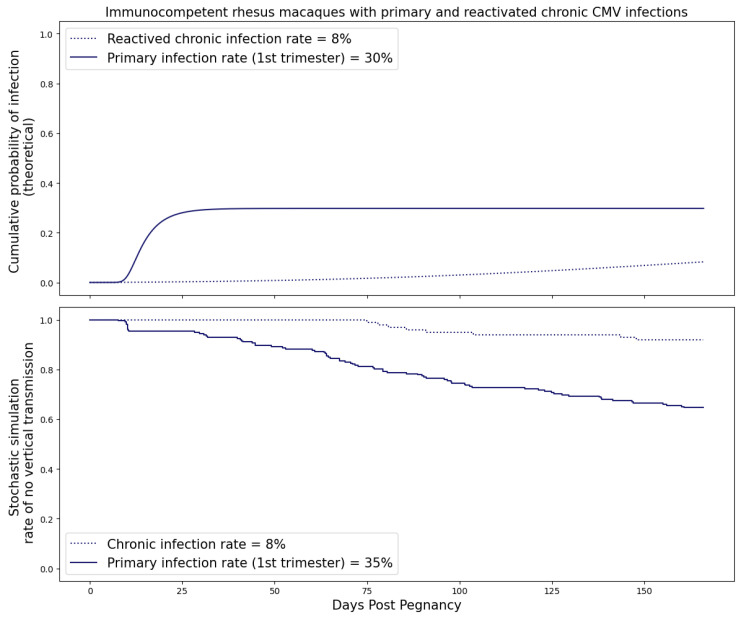
Probability of transplacental transmission of reactivated chronic RhCMV infections during different trimesters of pregnancy. The maternal dynamics come from the equilibrium values of the population fits of immunocompetent groups of rhesus macaques. Graph generated taking threshold M=500 copies. The blue dashed line refers to immunocompetent dams with reactivated chronic infection. The blue solid line refers to immunocompetent dams that are infected during the first trimester of pregnancy.

**Figure 12 viruses-15-02040-f012:**
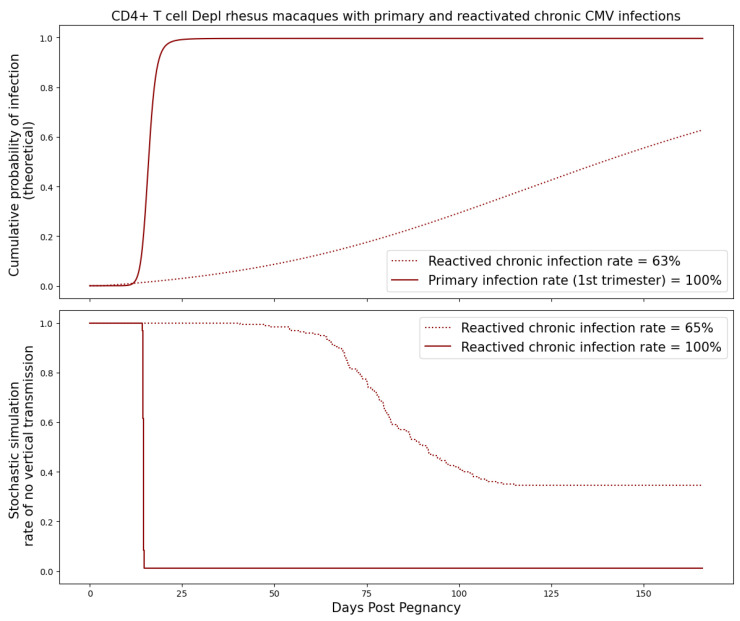
Probability of transplacental transmission of reactivated chronic maternal RhCMV infections during different trimesters of pregnancy. The maternal dynamics come from the equilibrium values of the population fits of CD4+ T-cell-depleted rhesus macaques. Graph generated taking threshold M=500 copies. The red dashed line refers to CD4+ T-cell-depleted dams with reactivated chronic infection. The red solid line refers to CD4+ T-cell-depleted dams that are infected during the first trimester of pregnancy.

**Table 1 viruses-15-02040-t001:** System of ordinary differential equation model parameter estimates.Note that “IC” stands for “immunocompetent” and “CD4+ Depl” stands for “CD4+ T-cell-depleted”.

Param	Description	IC	CD4+ Depl	Units	Ref [6]
ρ	Virion-induced immune response	2	3	cells/(virions · day)	5
*n*	Productivity of infected cell	$6\cdot 10^{1}$	$6\cdot 10^{1}$	cells/µL-blood	$5\cdot 10^{1}$
δ	Rate of viral-induced cell death	1.75	$2.5\cdot 10^{-1}$	day${}^{-1}$	$2\cdot 10^{-1}$
*c*	Rate of viral clearance	$1.5\cdot 10^{-1}$	$2.8\cdot 10^{-1}$	day${}^{-1}$	$3\cdot 10^{-1}$
*k*	Infection rate constant	$2.6\cdot 10^{-4}$	$4.5\cdot 10^{-4}$	µL/(virions · day)	$1\cdot 10^{-4}$
*m*	Immune-induced cell lysis	$8.3\cdot 10^{-2}$	$3.2\cdot 10^{-3}$	µL/(cells · day)	$1\cdot 10^{-1}$
$\alpha_{0}$	Exit and reactivation rate for monocytes	$1.7\cdot 10^{-2}$	$1\cdot 10^{-1}$	µL/(cells · day)	$2\cdot 10^{-1}$
κ	Rate of latency	$5.5\cdot 10^{-1}$	$1.4\cdot 10^{-3}$	µL/(cells · day)	$2\cdot 10^{-3}$
$\lambda_{E}$	Homeostatic replenishment of immune cells	$3.1\cdot 10^{-4}$	$4.5\cdot 10^{-4}$	day${}^{-1}$	$4\cdot 10^{-2}$
λ	Cell replenishment rate	$1\cdot 10^{-3}$	$7.1\cdot 10^{-4}$	day${}^{-1}$	$1\cdot 10^{-3}$
*e*	CMV-specific effector cell term	18	4	cells/µL-blood	9
$\epsilon_{S}$	Level of immune suppression	0	$9.9\cdot 10^{-1}$	-	0
*f*	Number of infecting virions per cell	1.4	$1\cdot 10^{-1}$	-	1
$r_{S}$	Equilibrium level of susceptible cells	$5.2\cdot 10^{2}$	$6\cdot 10^{2}$	cells/µL-blood	$4\cdot 10^{2}$
$r_{L}$	Equilibrium level of latent T cells	$7\cdot 10^{-2}$	$7\cdot 10^{-2}$	cells/µL-blood	$4\cdot 10^{-2}$
$t_{H}$	Half-life of virions during antiviral treatment	2	2	days	2
$t_{D}$	Doubling time of the virions	1.5	1.5	days	1.5
$\tilde{E}$	Equilibrium level of CMV-specific effector cells	10	10	cells/µL-blood	10
$\tilde{V}$	Equilibrium level of virions	$2\cdot 10^{-2}$	$2\cdot 10^{-2}$	virions/µL-blood	$2\cdot 10^{-2}$

**Table 2 viruses-15-02040-t002:** Stochastic simulation parameter estimates.

Parameter	Description	Value	Units	Ref
$S_{0}$	Initial susceptible cell population	$4\cdot 10^{8}$	cells	[12]
$L_{0}$	Initial latently infected cell population	1	cells	[12]
$I_{0}$	Initial infected cell population	0	cells	[12]
$V_{0}$	Initial viral population	0	virions	[12]
γ	Death rate of susceptible cells	1/4.5	cells/day	[12]
β	Rate of new infected cell	$3\cdot 10^{-12}$	1/(virions·cells·day)	[12]
α	Lag (days) between cell infection and viral growth	1	day	[12]
*d*	Death rate of infected cells	0.77	cells/day	[12]
ω	New viruses per infected cell per day	1600	1/(virions·day)	[12]
*c*	Natural decay of virus	2	virion/day	[12]

**Table 3 viruses-15-02040-t003:** Probability of transplacental transmission with HIG treatment for immunocompetent group.

HIG Infusion Day $T_{\mathrm{infusion}}$	Effectiveness $\epsilon_{\mathrm{eff}}$ = 100%	Effectiveness $\epsilon_{\mathrm{eff}}$ = 75%	Effectiveness $\epsilon_{\mathrm{eff}}$ = 50%
Not Treated	35%	35%	35%
Day 21	27%	30%	32%
Day 14	16%	22%	27%
Day 7	4%	14%	22%

**Table 4 viruses-15-02040-t004:** Probability of transplacental transmission with HIG treatment for the CD4+ T-cell-depleted group.

HIG Infusion Day $T_{\mathrm{infusion}}$	Effectiveness $\epsilon_{\mathrm{eff}}$ = 100%	Effectiveness $\epsilon_{\mathrm{eff}}$ = 75%	Effectiveness $\epsilon_{\mathrm{eff}}$ = 50%
Not Treated	100%	100%	100%
Day 21	99%	99%	100%
Day 14	86%	96%	99%
Day 7	32%	87%	97%

## Data Availability

Code and data used in this paper are available at https://github.com/yishu0524/CMV_placental-transmission (accessed on 28 September 2023).

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
