# Peer review of "Mathematical Modeling of Rhesus Cytomegalovirus Transplacental Transmission in Seronegative Rhesus Macaques"

_viruses, 2023, doi:10.3390/v15102040_

Round 1

Reviewer 1 Report

This paper describes mathematical modeling of CMV infection of pregnant rhesus macaque dams and fetal transmission. The experimental data used to inform the data are rare and valuable for providing insights into the viral dynamics in the mother, placenta and fetus with a granularity that is not possible in humans. Furthermore, the authors have access to experiments in which immune parameters were perturbed, either by maternal CD4 T cell depletion or passive immunization with immune globulin. Building on existing models in humans, placental and fetal compartments are added and fit to the RhCMV experimental data.

The subject matter is of great importance and furthers this potential for the RhCMV model to inform the biology of cCMV as well as the development of prevention strategies. Generally, the methods and conclusions appear sound, and certain findings are extremely provocative. These include the possibility that the association between higher risk of fetal infection and later gestational time is due to increasing greater placental surface, and the requirement of hyperimmune globulin to be extremely effective and given within 2 weeks of maternal infection to have a meaningful impact on congenital transmission. Nevertheless, certain aspects of the study methods and data could be better explained.

In the methods, it is stated that PCR reactions were considered positive if 2/6 plasma replicates or 2/12 amniotic fluid replicates were positive. What is the justification for these criteria?

Amniotic fluid data are mentioned multiple times but not shown. Though it is implied that they are in Fig 1, they are not. It is also not clear to what extent the longitudinal amniotic fluid data are used, as opposed to just ever/never positive as an indicator of transmission to the fetus. For example, were these data used to inform the time that virus arrives in the fetal (or placental) compartment? Were the viral loads over time in amniotic fluid fit to the model? Does the threshold M refer to a viral load in amniotic fluid? Were values of M<500 seen in amniotic fluid that did not result in fetal infection? Similarly, were placental PCR data exploited?

The modeling of reactivation is interesting, but this appears only to be reported for CD4-depleted dams. Does the model predict reactivation and vertical transmission among immunocompetent dams? What is the frequency of reactivation, and the likelihood of fetal infection per reactivation event, in immunocompetent and immunocompromised hosts? These are questions of great clinical interest.

The parameter ratio (r) is hard to understand from a biological perspective. A limitation of target cells seems highly unlikely in any compartment since this would seem to imply vast lytic replication and tissue destruction. In addition, CMV only becomes latent in very select cell types (e.g., myeloid progenitors) that would not be expected to represent a large proportion of the target cell population in any compartment. This should be better explained/justified both generally and in the sensitivity analyses.

It appears that additional data from this model have been generated by this grouo, including from seropositive dams that were reinfected (according to a bioRxiv preprint). Would it not be useful to exploit all the available data from this model?

Parts of the paper are quite awkwardly written, and the entire manuscript should be reviewed carefully for readability. For example:

"Cytomegalovirus, or CMV, causes the most common human congenital infection and can result in multiple long-term deficits in infants, especially neurological defects and hearing loss. Congenital CMV (cCMV) occurs in 0.7% of all pregnancies [1] and is the main cause of non-genetic hearing loss as well as permanent sensory nerve and neurocognitive impairment in infants worldwide." Much better would be something like, "Cytomegalovirus, or CMV, is the most common cause of congenital infection in humans and results in enormous childhood morbidity. Congenital CMV (cCMV) occurs in 0.7% of all pregnancies [1] and is a major cause of childhood hearing loss and neurodevelopmental delay." Also, it should be noted that a few sentences earlier, in the abstract, the estimate 1 in 200 infants (0.5%) is given.

"congenital infection of the fetus"

"Our transmission model makes predictions...  and generates testable predictions."

"The model was calibrated using data from primary RhCMV infection of seronegative Rhesus macaque dams [3,4]. In these experiments, pregnant RhCMV seronegative rhesus macaques were assigned to an immunocompetent group (n=15) or an immune-suppressed group (n=9)." The word seronegative in the first sentence is redundant, as primary infection always by definition occurs in a seronegative host and, furthermore, the next sentence explicitly states that they are seronegative. 

The verb tense is variably present and past: "We used the parameters for HCMV infection given in Kepler... When we perform parameter estimation..."

The abbreviation "cCMV" for congenital CMV infection is used inconsistently, and often "congenital CMV" (with the infection implied) is used instead.

The labels above the bottom 2 panels of Fig 6 are confusing or incorrect, and there is a typo in Fig 8 ("Days of gastation").

Author Response

Thank you very much for your suggestions, you can also view our response attachment in the Author's Notes File.

Reviewer 1: 

Comments and Suggestions for Authors: 

This paper describes mathematical modeling of CMV infection of pregnant rhesus macaque dams and fetal transmission. The experimental data used to inform the data are rare and valuable for providing insights into the viral dynamics in the mother, placenta and fetus with a granularity that is not possible in humans. Furthermore, the authors have access to experiments in which immune parameters were perturbed, either by maternal CD4 T cell depletion or passive immunization with immune globulin. Building on existing models in humans, placental and fetal compartments are added and fit to the RhCMV experimental data. 

The subject matter is of great importance and furthers this potential for the RhCMV model to inform the biology of cCMV as well as the development of prevention strategies. 

Generally, the methods and conclusions appear sound, and certain findings are extremely provocative. These include the possibility that the association between higher risk of fetal infection and later gestational time is due to increasing greater placental surface, and the requirement of hyperimmune globulin to be extremely effective and given within 2 weeks of maternal infection to have a meaningful impact on congenital transmission. 

We deeply appreciate the reviewer’s comments on the value of our modeling efforts and provide our responses to specific points raised below. 

Nevertheless, certain aspects of the study methods and data could be better explained. 

In the methods, it is stated that PCR reactions were considered positive if 2/6 plasma replicates or 2/12 amniotic fluid replicates were positive. What is the justification for these criteria? 

We were concerned that a single positive sample could have been due to contamination but believe that 2 or more positive samples was likely to indicate a true positive even for qPCR levels just above the LOD. We have clarified this in the text on Line 71 which now reads “To reduce the risk of false positives from inadvertent contamination, when the qPCR results approached this lower limit of detection, PCR reactions were deemed positive if at least 2 out of 6 replicates in the plasma or 2 out of 12 replicates in the amniotic fluid showed positive results.”.  

Amniotic fluid data are mentioned multiple times but not shown. Though it is implied that they are in Fig 1, they are not. It is also not clear to what extent the longitudinal amniotic fluid data are used, as opposed to just ever/never positive as an indicator of transmission to the fetus. For example, were these data used to inform the time that virus arrives in the fetal (or placental) compartment? Were the viral loads over time in amniotic fluid fit to the model? Does the threshold M refer to a viral load in amniotic fluid? Were values of M<500 seen in amniotic fluid that did not result in fetal infection? Similarly, were placental PCR data exploited? 

PCR positivity from amniotic fluid samples was used to provide an upper bound of time to the first fetal infection event, and this was used to calibrate the parameters for the placental transmission model. Relatively few animals (6 out of 15) had amniotic fluid positivity, and of those that did, this was often transient or even only seen at a single time point.  Given the small sample size and noisiness of the data, we did not believe that it was reasonable to model viral loads over time in the amniotic fluid. Because of this, we believe that the M should be interpreted as an (unobserved) viral load in the fetal blood circulation. We use amniotic fluid positive as the indicator of infection in our model to match what is used to diagnose CMV infection in human pregnancies. Placental PCR data was not used for model calibration as the viral loads in the placenta were not measured spatially and were only available at a single time point after delivery. From our unpublished data (manuscript in preparation), placental PCR and amniotic fluid positivity were not fully concordant — in an analysis of 12 of the rhesus macaques where placental tissue was available, only 3 out of the animals that were amniotic fluid positive were also placenta positive, while 4 of the animals that were placenta positive were not amniotic fluid positive. Only a single fetus was tissue positive; this fetus was both amniotic fluid and placenta positive. This may reflect temporal sampling issues with the amniotic fluid or placental tissue but is generally consistent with the model assumptions that there is viral attrition during placental transmission, and that infection after placental transmission is a low probability stochastic process for any individual virion.  

The modeling of reactivation is interesting, but this appears only to be reported for CD4- depleted dams. Does the model predict reactivation and vertical transmission among immunocompetent dams? What is the frequency of reactivation, and the likelihood of fetal infection per reactivation event, in immunocompetent and immunocompromised hosts? 

These are questions of great clinical interest. 

Vertical transmission due to reactivation in immunocompetent dams is not fully addressable by the current model, since the viral load (VL) in the dam is modeled using deterministic ODEs. In the absence of immunosuppression, the long-term VL will stay at the same level and there is no reactivation in the sense of a “blip” in VL. The model thus predicts that the likelihood of vertical transmission among immunocompetent dams is negligible since maternal VL is continually suppressed by the immune system. However, vertical transmission events in immunocompetent mothers could also result from reinfection, potentially with a different strain for which there is less immunity, and/or if pregnancy itself is a state of relative immunodeficiency. To model reactivation events and the per-event risk of transmission in immunocompetent dams, we would have to use a stochastic model that allows random reactivation events, for example, as a Poisson or Hawkes process. Since this requires the construction and calibration of a new model, we believe that this is beyond the scope of the current manuscript.  

The parameter ratio (r) is hard to understand from a biological perspective. A limitation of target cells seems highly unlikely in any compartment since this would seem to imply vast lytic replication and tissue destruction. In addition, CMV only becomes latent in very select cell types (e.g., myeloid progenitors) that would not be expected to represent a large proportion of the target cell population in any compartment. This should be better explained/justified  

both generally and in the sensitivity analyses. 

The parameter r is only used for the stochastic model in the fetus. That model does not assume any target cell limitation; in fact, since we only model infection of cells up to the threshold M, target cell limitation is irrelevant. While CML becomes latent only in select cell types such as myeloid cells, the model assumes that initial infection occurs because a virion coming across the placental barrier infects a blood cell in the fetal circulation, which is most likely myeloid and hence permissive to infection. We also assume that initial rounds of infection may also infect other myeloid cells in the blood before parenchymal cell infection. Hence it does not seem too unreasonable to assume that half of all virions in early infection enter a latent stage. However, we acknowledge that the exact value of r is highly imprecise, and hence provide sensitivity analysis that include the extremes of r = 0 and r = 1. We have expanded our rationale for modeling r in the text between Lines 158 and 171. 

It appears that additional data from this model have been generated by this group, including from seropositive dams that were reinfected (according to a bioRxiv preprint). Would it not be useful to exploit all the available data from this model? 

In these experiments (BioRχiv preprint Protective effect of pre-existing natural immunity in a nonhuman primate reinfection model of congenital cytomegalovirus infection at https://doi.org/10.1101/2023.04.10.536057), of 5 seropositive CD4 T cell-depleted dams (i.e., with pre-existing immunity) reinfected with CMV, only 1 showed placental transmission and there were no adverse fetal sequelae — in contrast with primary infection in CD4 T cell-depleted dams where there is 100% transmission and often fetal loss — suggesting that preconception maternal CMV-specific CD8+ T lymphocyte and/or humoral immunity can protect against cCMV infection. To accurately model reinfection in seropositive dams under CD4 T cell-depletion as in these experiments, we would need a new maternal and possibly placental model that explicitly includes both CD4 and non-CD4 immune effector (e.g., memory CMV-specific CD8 T cells) mechanisms. This would be a more complex model with little data available for model calibration; n=5 with only 1 transmission event. While this is an important line or research that we would like to pursue in the future, it would require the development of a new model, and we believe that it is outside the scope of this manuscript. With the existing model, if we make the simplifying assumption that in dams with pre-existing immunity, CD4 T cell -depletion leads to less immune suppression than in primary infection, simulations suggest that reinfection can be controlled leading to a small risk of vertical transmission. For example, in the following figure, we introduced 100 copies of CMV virus/mL into the system, and changed the immune suppressed level to 0.5, there is no sustained infection. Given the highly speculative nature of this result and the paucity of experimental data to estimate the likely level of immune suppression, we are hesitant to include these results in the manuscript. (Please see the figure in Author's Notes File)

.  

Comments on the Quality of English Language: 

Parts of the paper are quite awkwardly written, and the entire manuscript should be reviewed carefully for readability. For example: 

"Cytomegalovirus, or CMV, causes the most common human congenital infection and can result in multiple long-term deficits in infants, especially neurological defects and hearing loss. Congenital CMV (cCMV) occurs in 0.7% of all pregnancies [1] and is the main cause of non-genetic hearing loss as well as permanent sensory nerve and neurocognitive impairment in infants worldwide." Much better would be something like, "Cytomegalovirus, or CMV, is the most common cause of congenital infection in humans and results in enormous childhood morbidity. Congenital CMV (cCMV) occurs in 0.7% of all pregnancies [1] and is a major cause of childhood hearing loss and neurodevelopmental delay." Also, it should be noted that a few sentences earlier, in the abstract, the estimate 1 in 200 infants (0.5%) is given. 

Change to the suggested wording “Cytomegalovirus, or CMV, is the most common cause of congenital infection in humans and results in enormous childhood morbidity. Congenital CMV (cCMV) occurs in 0.7% of all pregnancies [1] and is a major cause of childhood hearing loss and neurodevelopmental delay”, change is reflected In line 14 

Abstract changed to “Approximated 0.7% of the infants ….” to be consistent. Change is reflected in line 1. 

"congenital infection of the fetus" 

Changed to “congenital infection in the fetus”. Change is reflected in line 52. 

"Our transmission model makes predictions... and generates testable predictions." "The model was calibrated using data from primary RhCMV infection of seronegative  

Rhesus macaque dams [3,4]. In these experiments, pregnant RhCMV seronegative rhesus  

macaques were assigned to an immunocompetent group (n=15) or an immune-suppressed group (n=9)." The word seronegative in the first sentence is redundant, as primary infection always by definition occurs in a seronegative host and, furthermore, the next sentence explicitly states that they are seronegative. 

The word “seronegative” in the first sentence is removed. Change is reflected in line 62. 

The verb tense is variably present and past: "We used the parameters for HCMV infection given in Kepler... When we perform parameter estimation..." 

Verb tense describing our action has been changed to past for consistency. 

The abbreviation "cCMV" for congenital CMV infection is used inconsistently, and often "congenital CMV" (with the infection implied) is used instead. 

“congenital CMV (cCMV)” is used in abstract and introduction. All later occurrences have been changed to cCMV. 

The labels above the bottom 2 panels of Fig 6 are confusing or incorrect, and there is a typo in Fig 8 ("Days of gastation"). 

Figure 6 label has been fixed.  

Figure 8 typo has been fixed. 

Reviewer 2 Report

The authors considered and validated a mathematical model for the transmission of CMV in the placenta. The model includes an ODE model for the immune and viral dynamics in the dam, a diffusion model for the transmission of CMV across the placenta, and stochastic processes for the arrival and evolution of CMV virus in the fetus. These modeling choices, in my opinion, are very relevant and suitable for the types of questions the authors considered in the paper. Overall, the paper is very well written, and the results are promising and are correlated well with clinical observations. Definitely, the work is very interesting and can have implications in future CMV studies as well as optimal design of NHP intervention experiments. Other comments:

(1) It is noted that a sensitivity analysis was performed to study the uncertainty of the CMV life cycle in the fetus, would a sensitivity analysis on the ODE model for the CMV dynamics in the dam help to improve the fit of the model to the experimental data?

(2) How do you justify using the diffusion coefficient of HIV through stroma as the diffusion coefficient in your model?

(3) Two stochastic models were considered for the transmission of CMV in the fetus. The choice of a specific model was based on a uniform random variable s between 0 and 1 and a threshold value r. How do you justify this strategy biologically? Ditto for the choice of r = 0.5.

(4) Some discussions should be provided on how biologically relevant or reasonable the fitted parameters are in Table 1. 

Author Response

Thank you very much for your suggestions, you can also view our response attachment in the Author's Notes File.

Reviewer 2: 

Comments and Suggestions for Authors: 

The authors considered and validated a mathematical model for the transmission of CMV in the placenta. The model includes an ODE model for the immune and viral dynamics in the dam, a diffusion model for the transmission of CMV across the placenta, and stochastic processes for the arrival and evolution of CMV virus in the fetus. These modeling choices, in my opinion, are very relevant and suitable for the types of questions the authors considered in the paper. Overall, the paper is very well written, and the results are promising and are correlated well with clinical observations. Definitely, the work is very interesting and can have implications in future CMV studies as well as optimal design of NHP intervention experiments. Other comments: 

It is noted that a sensitivity analysis was performed to study the uncertainty of the CMV life cycle in the fetus. Would a sensitivity analysis on the ODE model for the CMV dynamics in the dam help to improve the fit of the model to the experimental data? 

To diagnose the ODE fit, we include several figures in Appendix A. Specifically, the Visual Predictive Check (VPC) analysis in Appendix A Figure 4 demonstrates the alignment of observed and predicted percentiles, all falling within their respective prediction intervals. This evaluation assesses the model's accuracy in capturing both central trends and variability across time intervals. The agreement between empirical and theoretical percentiles suggests a robust fit of our model to the data. 

How do you justify using the diffusion coefficient of HIV through stroma as the diffusion coefficient in your model? 

The reviewer is of course right that the diffusion coefficients of HIV though stroma and CMV through placenta could be different. Unfortunately, as far as we know, the diffusion coefficient of CMV in any tissue is not available, necessitating the use of the admittedly imperfect analogous HIV diffusion coefficient. For this reason, we have provided a sensitivity analysis in the manuscript in Section 3.5 where D is varied over a 4-fold range. 

Two stochastic models were considered for the transmission of CMV in the fetus. The choice of a specific model was based on a uniform random variable s between 0 and 1 and a threshold value r. How do you justify this strategy biologically? Ditto for the choice of r = 0.5. 

While CMV becomes latent only in select cell types such as myeloid cells, the model assumes that initial infection occurs because a virion coming across the placental barrier infects a blood cell in the fetal circulation, which is most likely myeloid and hence permissive to infection. We also assume that initial rounds of infection may also infect other myeloid cells in the blood before parenchymal cell infection. Hence it does not seem too unreasonable to assume that half of all virions in early infection enter a latent stage. However, we acknowledge that the exact value of r is highly imprecise, and hence provide sensitivity analysis that include the extremes of r = 0 and r = 1. We have expanded our rationale for modeling r  in the text on Lines 158 and 171. 

Some discussions should be provided on how biologically relevant or reasonable the fitted parameters are in Table 1.  

Justifications for these values are provided in the original paper with respect to pertinent clinical measurements in Section 3, and the approximation of the parameters in Section 4. In the table below (expansion of Table 1 in the manuscript), we show our fitted parameter values for both immunocompetent and CD4 T cell-depleted animals together with the values in the original paper. While some variation is expected since we are comparing parameter values calibrated in humans versus macaques, our fitted values do not deviate excessively from those derived in the Kepler 2009 paper. We have replaced the ordinal Table 1 with this revised Table (please see the table in the Author's Notes File), and added commentary on parameter estimates on Lines 181-183. 

Reviewer 3 Report

The authors present an interesting and important study involving CMV and infection in Rhesus Macaques with the intent to relate this to human infection.  I have several issues that I believe should be addressed however:

1. gestational development involves parallel, non aligned organ development and should be evaluated at a higher resolution than "trimester"....rather examining gestational week

2. a comparison of the developmental stages of the fetus in the Macaque and Human need to be compared to support potential application of the results at a high level of consistency

3. the authors note the ongoing placental development during gestation and should also consider maternal development as well so that fetus, placenta and mother are accurately analyzed in terms of their state of development at time of infection, etc for a more targeted comparison

4. while consideration of the growth curve of the placenta is included, it is also relevant to discuss pathophysiologic changes that are also occurring in the placenta and mother

Author Response

Thank you very much for your suggestions, you can also view our response attachment in the Author's Notes File.

Reviewer 3: 

Comments and Suggestions for Authors: 

The authors present an interesting and important study involving CMV and infection in Rhesus Macaques with the intent to relate this to human infection. I have several issues that I believe should be addressed however: 

gestational development involves parallel, non aligned organ development and should be evaluated at a higher resolution than "trimester" rather examining gestational week 

We appreciate the subtle insight pointed out by the reviewer that all organ systems do not develop according to the same timetable and trimesters only provide a high level approximation for transitions such as the hematopoietic system (specifically which and when in relation to transition from fetal liver to bone marrow and other tissues like the spleen and thymus) as well as organs such as the lung or the brain that go through a variety of stages with critical cells at a particular time. However, mathematical models are abstractions and make simplifying assumptions so we can rigorously examine the consequences of these assumptions. Just as important, mathematical models need to be calibrated with existing experimental data to make predictions, limiting the complexity of the models we can build. While we cannot build and calibrate models with biologically realistic organ development schedules for these reasons, we note that the model can predict outcomes for infection at any gestational age (e.g., figures 8 and 9) and that the trimester risk predictions were averaged over the smoothed curves. 

a comparison of the developmental stages of the fetus in the Macaque and Human need to be compared to support potential application of the results at a high level of consistency. 

In the conclusion on Lines 365-369, we discuss the impact of potential differences in developmental stages and refer the reader to Nakamura et al — Non-human primates as a model for human development (PMC8185448) — for a study of these issues. As discussed in the response above, a model with detailed developmental schedule for various organ systems cannot be meaningfully modeled with currently available data and hence is not attempted. However, we believe that the fundamental conservation of developmental processes across humans and monkeys, as pointed out in Nakamura et al, argues for the plausibility of using the model to make plausible inferences about human pregnancies for hypothesis generation.  

the authors note the ongoing placental development during gestation and should also consider maternal development as well so that fetus, placenta and mother are accurately analyzed in terms of their state of development at time of infection, etc for a more targeted comparison … while consideration of the growth curve of the placenta is included, it is also relevant to discuss pathophysiologic changes that are also occurring in the placenta and mother 

Apart from placenta growth, several developmental processes in mother, placenta, or fetus can potentially affect the risk of placental transmission, including changes in concentration of immune effector cells in the mother, transplacental trafficking of IgG vs IgM, or change in number of cells susceptible to CMV infection in the developing fetus, etc. Unfortunately, a detailed model incorporating all these effects cannot be meaningfully calibrated with existing data. To be clear, we are only claiming that the model generates the novel hypothesis that placental growth itself could be an unsuspected major determinant of the transmission risk across different stages of pregnancy. For context, we have now included a discussion of other pathophysiologic changes that could impact the risk of CMV transmission across different pregnancy stages on Lines 345-351.   

Round 2

Reviewer 1 Report

Yes, the authors have addressed my comments adequately and I would suggest accepting the manuscript now.